# The effect of clinically elevated body mass index on physiological stress during manual lifting activities

Sergio A. Lemus[1], Mallory Volz[2], Eduard Tiozzo[3]*, Arlette Perry[4,5,6], Thomas M. Best[2,7,8], Francesco Travascio[1,7,9]*

1 Department of Mechanical and Aerospace Engineering, University of Miami, Coral Gables, FL, United States of America, 2 Department of Biomedical Engineering, University of Miami, Coral Gables, FL, United States of America, 3 Department of Physical Medicine and Rehabilitation, University of Miami, Miami, FL, United States of America, 4 Department of Kinesiology and Sport Sciences, University of Miami, Coral Gables, FL, United States of America, 5 Laboratory of Clinical and Applied Physiology, University of Miami, Coral Gables, FL, United States of America, 6 School of Education and Human Development, University of Miami, Coral Gables, FL, United States of America, 7 Department of Orthopaedics, University of Miami, Miami, FL, United States of America, 8 UHealth Sports Medicine Institute, Coral Gables, FL, United States of America, 9 Max Biedermann Institute for Biomechanics at Mount Sinai Medical Center, Miami Beach, FL, United States of America

* f.travascio@miami.edu (FT); e.tiozzo@med.miami.edu (ET)

**Data Availability Statement:** All relevant data are within the paper and its Supporting Information file.

## Abstract

Individuals with a body mass index (BMI) classified as obesity constitute 27.7% of U.S. workers. These individuals are more likely to experience work-related injuries. However, ergonomists still design work tasks based on the general population and normal body weight. This is particularly true for manual lifting tasks and the calculation of recommended weight limits (RWL) as per National Institute of Occupational Safety & Health (NIOSH) guidelines. This study investigates the effects of BMI on indicators of physiological stress. It was hypothesized that, for clinically elevated BMI individuals, repeated manual lifting at RWL would produce physiological stress above safety limits. A repetitive box lifting task was designed to measure metabolic parameters: volume of carbon dioxide ($VCO_2$) and oxygen ($VO_2$), respiratory exchange ratio (RER), heart rate (HR), and energy expenditure rate (EER). A two-way ANOVA compared metabolic variables with BMI classification and gender, and linear regressions investigated BMI correlations. Results showed that BMI classification represented a significant effect for four parameters: $VCO_2$ ($p < 0.001$), $VO_2$ ($p < 0.001$), HR ($p = 0.012$), and EER ($p < 0.001$). In contrast, gender only had a significant effect on $VO_2$ ($p = 0.014$) and EER ($p = 0.017$). Furthermore, significant positive relationships were found between BMI and $VCO_2$ ($R^2 = 59.65\%$, $p < 0.001$), $VO_2$ ($R^2 = 45.01\%$, $p < 0.001$), HR ($R^2 = 21.86\%$, $p = 0.009$), and EER ($R^2 = 50.83\%$, $p < 0.001$). Importantly, 80% of obese subjects exceeded the EER safety limit of 4.7 kcal/min indicated by NIOSH. Indicators of physiological stress are increased in clinically elevated BMI groups and appear capable of putting these individuals at increased risk for workplace injury.

**Funding:** Drs. Travascio and Tiozzo received funds from The University of Miami (award UM-PRA-2022-3243) to complete this study. The funders had no role in study design, data collection and analysis, decision to publish, or preparation of the manuscript.

**Competing interests:** The authors have declared that no competing interests exist.

# 1. Introduction

In most occupational and industry groups, the prevalence of injury is positively correlated to the proportion of high body mass index (BMI) individuals comprising the workforce [1–3]. Overall, 27.7% of U.S. workers meet the BMI criterion for obesity [4]. These obese workers are up to 50% more likely to experience work related injuries, when compared to average BMI individuals [5]. Numerous causal and mechanistic relationships between the clinically elevated BMI population and increased risk of injury have been proposed [6]. For example, obesity is associated with a decrease in capillary density and reduced blood flow to skeletal muscle [7], limiting the supply of oxygen and nutrients to working tissue. This can potentially result in a faster onset of muscle fatigue during sustained exertions [8]. In addition, the muscular fiber composition of obese people has been shown to have a lower percentage of fatigue resistant fibers when compared to average BMI individuals [9]. It is also already established that obese people have a higher cardiovascular demand when performing basic occupational tasks, due to reduced metabolic and conditioning performance [10–13].

Despite the adverse effects of clinically elevated BMI on the prevalence of musculoskeletal injuries in blue collar workers, workspace and work tasks are still being designed using anthropometric characteristics derived from average "healthy" workers [14, 15]. Such is the case for manual lifting tasks that account for a major portion of activities in manual work occupations [16]. Sound ergonomic practice suggests that such tasks should be designed based on the revised National Institute for Occupational Safety and Health (NIOSH) lifting equation (RNLE) [17]. Specifically, the RNLE is an ergonomic tool designed to reduce the physical stress in manual lifting activities [17]. Developed on the basis of both physiological and biomechanical safety criteria, the RNLE calculates recommended weight limits (RWL) with the purpose of safely regulating the energy expenditure and musculoskeletal loads during manual lifting [18]. The RNLE comprises experimental coefficients (multipliers) accounting for work geometry (e.g., initial. and final location of the object to be lifted, etc.), frequency, and duration of the task. According to the RNLE, the RWL is modulated according to the frequency and duration of the lifting task in order maintain a person's energy expenditure within a safety limit of 4.7 kcal/min [18]. Furthermore, the RNLE multipliers are based on experimental data obtained from an average BMI population [14, 15]. In other words, the calculation of RWL does not explicitly account for effects of BMI. Consequently, we hypothesized that, for large BMI individuals, the execution of manual lifting at RWL would produce a level of physiological stress above safety limits. Therefore, the aim of this study was to measure and compare metabolic parameters indicative of physiological stress in normal weight, overweight, and obese individuals while performing manual lifting activities at RWL.

# 2. Methods

A repetitive box lifting task was designed to measure metabolic outputs and explore the effects of BMI on physiological stress. Subjects lifted a box from the floor to the table and back at a predetermined frequency while metabolic measurements were conducted. Subject recruitment, work geometry, and detailed experimental procedures are described below.

## 2.1 Subjects recruitment

The methods used in this cross-sectional study were approved by the Internal Review Board of the University of Miami (IRB ID: 20211175). All participants were informed of the study procedures, provided written consent prior to beginning the experiment, and were given the option to withdraw at any time. Participants were recruited from the Miami-Dade metropolitan area through advertisement flyers posted in common areas on and surrounding the

University's main campus, as well as Facebook posts in local groups. Inclusion criteria consisted of any adult with no current or prior history of musculoskeletal injury, or any medical condition preventing him/her from performing the physical activity planned in the experiments. Exclusion criteria for the study consisted of pregnancy, history of musculoskeletal injuries in the past 6 months or any chronic cardiopulmonary-related conditions that may be exacerbated during tests (including but not limited to asthma, chronic obstructive pulmonary disease, congestive heart failure, and/or lung disease). Total enrollment was 30 people stratified into groups of 10 (5 males and 5 females with gender being self-reported) for 3 BMI classifications based on the definitions of the World Health Organization [19]: normal weight ($18.5 \leq BMI \leq 24.9$), overweight ($25 \leq BMI \leq 29.9$), and obese ($BMI \geq 30$). A single subject with a BMI of 16.71 was determined to not be statistically different from the metabolic measures of the normal classification and thus was grouped as such. Ethnic and racial composition of subjects was chosen to represent, as close as possible, Miami-Dade County demographics [20].

## 2.2 Work geometry and lifting task

Subjects stood in front of a box (10"x10"x10") and moved it from the floor to the 30" high table, and from the table to the floor. The box was lifted at a frequency of six lifts per minute for 20 minutes, which is considered a moderate exercise intensity based on the threshold guidelines for five intensity zones (very light, light, moderate, vigorous, near maximal/maximal) [21]. The hand grasps of the box were 10" away from the midpoint of the subject's ankles and 5" from the floor. A 'freestyle' lifting technique was allowed. The geometrical setting of the task is shown in Fig 1. The weight of the box was chosen to be equivalent to the RWL as defined in the revised NIOSH lifting equation RNLE [17]:

$$RWL = LC \times HM \times VM \times DM \times AM \times FM \times CM,$$

Where LC is a load constant equal to 51 pounds; the multipliers HM, VM and DM account for the horizontal and vertical location of the load with respect to the operator, and the vertical distance the load during lifting; the multiplier AM accounts for trunk rotation while lifting (asymmetry); FM varies according to frequency and duration of the lifting task; CM accounts for the hand-to-object coupling [17]. The specific numerical values of the RNLE multipliers, depending on the specific work geometry and the frequency of lifting, are reported in Table 1. The resulting RWL for the specific task investigated was 27.5 pounds.

## 2.3 Metabolic measurements

Metabolic data during the lifting task were acquired via a CardioCoachCO2 metabolic cart (KORR, Utah, USA) in conjunction with a heart rate monitor (Polar, Kempele, Finland). After a standardized automatic calibration, the equipment measured the volume of carbon dioxide production (VCO2), volume of oxygen consumption (VO2), and heart rate (HR). The values of VO2 and VCO2 were used to compute the respiratory exchange ratio (RER), which refers to the ratio of $CO_2$ production to $O_2$ consumption and indirectly shows skeletal muscle's oxidative capacity to obtain energy [22]:

$$RER = \frac{VCO_2}{VO_2}. \tag{1}$$

In addition, the rates of oxygen consumption (VO2) and carbon dioxide production (VCO2) were used to determine the energy expenditure rate (EER) according to the Weir

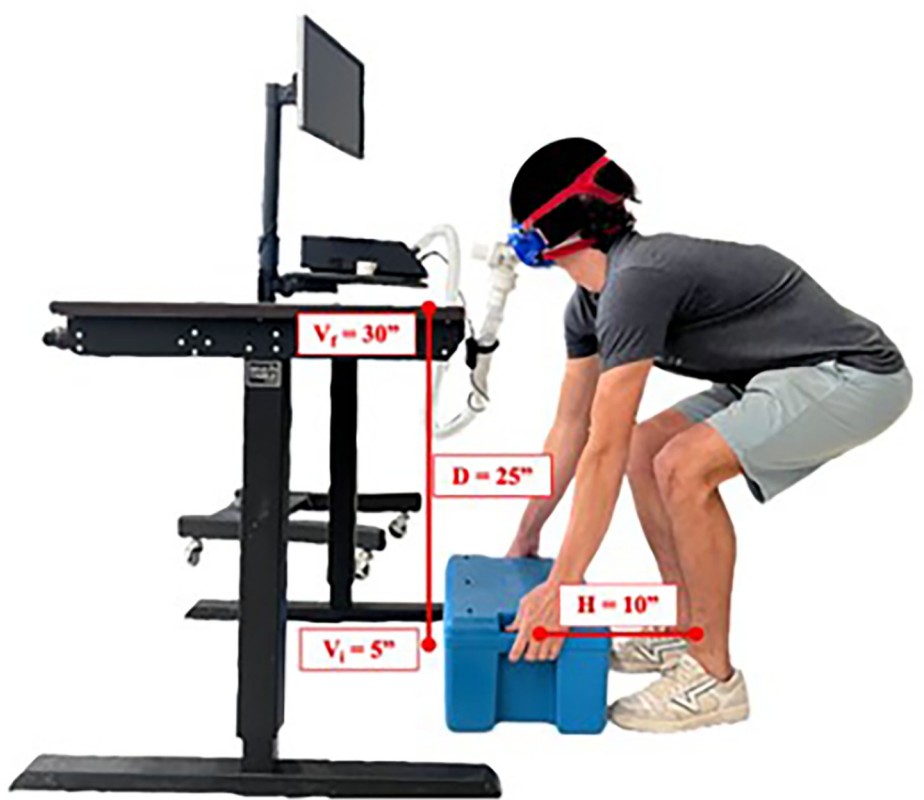

**Fig 1. Geometrical setting for experiment.** Subjects stood in front of a 27.5 pound box (10"x10"x10") and moved it from the floor to the 30" high table, and from the table to the floor at a frequency of six lifts per minute for 20 minutes. The hand grasps of the box were 10" away from the midpoint of the subject's ankles and 5" from the floor. A 'freestyle' lifting technique was allowed. The subject wore a mask that connected to the CardioCoachCO2 metabolic cart to measure VCO2 and VO2 and a heart rate monitor for the duration of the lifting activity.

equation [23]:

$$EER\left[\frac{kcal}{min}\right] = 3.94 \cdot VO_2\left[\frac{ml}{min}\right] + 1.11 \cdot VCO_2\left[\frac{ml}{min}\right]$$

(2)

Where the coefficients 3.94 and 1.11 relate the volume of gas exchange to the calories consumed per minute.

**Table 1. Summary of RNLE component.** The RNLE computes the RWL for a manual lifting task based on a specific variable that reflects the work geometry and lifting frequency of that particular task. These input values correspond to a RLNE multiplier, which is directly used in the RWL calculation [17].

| RNLE: *RWL = LC × HM × VM × DM × AM × FM × CM* | | | |
|---|---|---|---|
| **Variable** | **Input Value** | **RNLE Component** | **RWL** |
| *Load Constant (LC)* | | 51 lbs. | 27.5 lbs |
| *Horizontal Multiplier (HM)* | 10" | 1 | |
| *Vertical Multiplier (VM)* | 5" | 0.81 | |
| *Distance Multiplier (DM)* | 25" | 0.89 | |
| *Asymmetry Multiplier (AM)* | No trunk rotation | 1 | |
| *Frequency Multiplier (FM)* | 6 lifts/min over 20 min | 0.75 | |
| *Coupling Multiplier (CM)* | Good | 1 | |

## 2.4 Statistical analysis

Time-dependent measurements for VCO2, VO2, RER, HR, and EER for each BMI group were reported with sampling at 15 second intervals (Excel 2207, Microsoft Corporation, Washington, USA, and MATLAB (R2022a, MathWorks Inc., Massachusetts, USA). A five-point moving average was used to smooth the data points. When investigating statistical differences across metabolic parameters, the values of VCO2, VO2, RER, HR, and EER were those attained at steady state, according to the recommendations of indirect calorimetry published by The Academy of Nutrition and Dietetics (AND): after a 5-minute stabilization period, the steady state was determined when the coefficient of variation was less than 10% for all the metabolic parameters [24]. It has been reported that, at moderate exercise, steady state can be reached after 10–15 minutes [25–27]. In this study, the timeframe of observation was 20 minutes to guarantee full attainment of steady-state conditions. For all the subjects groups investigated, the metabolic parameters went through a transient period, varying from 5 to 10 minutes, in which their magnitudes changed (increasing for $VCO_2$, $VO_2$, HR and EER, and decreasing for RER); the final 5–10 minutes of the observational period were characterized by a steady state. The subsequent statistical analysis was conducted in Minitab® (21.1.1, Minitab LLC, Pennsylvania, USA). A two-way ANOVA followed by Fisher Pairwise comparison was performed to compare the steady-state values of the metabolic variables, with factors being the BMI classification (three levels) and gender (two levels). Additionally, simple linear regression models investigated the correlation of BMI with steady-state values of $VCO_2$, $VO_2$, RER, HR, and EER. Grubb's tests were performed to detect outliers. For all the statistical analyses performed, significance was set at 95% ($\alpha = 0.05$). Finally, a post-hoc power analysis was conducted using G*Power version 3.1.9.7 [28] to find the statistical power for metabolic outputs considering BMI and gender as main effects.

## 3. Results

The sample (n = 30) comprised of 50% males (n = 15) and 50% females (n = 15) with a mean age of 32.5 (SD = 12.3) and BMI of 28.5 (SD = 7.43), see Table 2. The overall racial/ethnic distributions of the subjects were as follows: 40.0% Hispanic/Latino (n = 12), 16.6% white, not Hispanic/Latino (n = 5), 10% black, or African American (n = 3), 16.6% Asian (n = 5), and 16.6% two or more races/ethnicities (n = 5), see Table 3. The overall workforce distribution of the subjects was as follows: 53% education (n = 16), 20% service (n = 6), 20% management and sales (n = 6), and 7% healthcare (n = 2), see Table 4.

The values of the metabolic parameters (VCO2, VO2, RER, HR, and EER) across the entire observational period of 20 minutes are reported in Fig 2. Data show that the magnitudes of

**Table 2. Age and anthropometric data of participants.** All the data are reported as mean ± standard deviation.

| | | Age [y.o.] | BMI [kg/m$^2$] |
|---|---|---|---|
| Normal BMI (n = 10) | Male (n = 5) | 28.2 ± 8.9 | 23.0 ± 1.1 |
| | Female (n = 5) | 31.2 ± 12.7 | 22.0 ± 2.7 |
| | Subtotal | 29.7 ± 11.1 | 22.5 ± 2.1 |
| Overweight BMI (n = 10) | Male (n = 5) | 32.8 ± 12.8 | 26.6 ± 1.5 |
| | Female (n = 5) | 42.2 ± 11.4 | 27.2 ± 0.7 |
| | Subtotal | 37.5 ± 13.0 | 26.9 ± 1.2 |
| Obese BMI (n = 10) | Male (n = 5) | 26.2 ± 4.9 | 40.5 ± 9.3 |
| | Female (n = 5) | 34.6 ± 13.8 | 31.5 ± 1.3 |
| | Subtotal | 30.4 ± 11.2 | 36.1 ± 8.0 |

**Table 3. Summary of subjects' race and ethnicity.** All the data are reported as number of subjects and percentages based on the total (n = 30). Data are compared to Miami-Dade County demographics as per 2020 Census.

| | Hispanic or Latino | White alone, not Hispanic or Latino | Black or African American | Asian | Two or more races |
|---|---|---|---|---|---|
| **Male (n = 15)** | 6 (20.0%) | 3 (10.0%) | 1 (3.3%) | 4 (13.3%) | 1 (3.3%) |
| **Female (n = 15)** | 6 (20.0%) | 2 (6.7%) | 2 (6.7%) | 1 (3.3%) | 4 (13.3%) |
| **Subtotal** | 12 (40.0%) | 5 (16.6%) | 3 (10%) | 5 (16.6%) | 5 (16.6%) |
| **Miami-Dade County 2020 Census [20]** | *69.1%* | *13.6%* | *17.4%* | *1.6%* | *1.3%* |

RER were similar across all BMI groups, see Fig 2C. In contrast, the general trend of the other metabolic parameters indicated that obese and overweight subjects experienced higher values of $VCO_2$, $VO_2$, HR and EER, when compared to the normal BMI participants. Importantly, after the first 5 minutes of lifting, the values of EER of normal (3.8 kcal/min) and overweight (4.3 kcal/min) subjects remained below the NIOSH safety limit (4.7 kcal/min), while that of the obese reached 5.6 kcal/min, surpassing the safety threshold by 0.9 kcal/min (19%).

Statistical comparisons of magnitudes of metabolic parameters at steady state across gender and BMI groups are reported in Table 5. First, for each parameter investigated, no significant interaction between BMI classification and gender was found. However, gender produced a significant main effect for VO2 (p = 0.014) and EER (p = 0.017): the magnitudes of these parameters found in men were higher than those of women. Post-hoc analysis showed that the power in determining statistical differences between genders for both VO2 and EER was larger than 95%. Also, BMI classification was a significant main effect for the four metabolic parameters VCO2 (p < 0.001), VO2 (p < 0.001), HR (p = 0.012) and EER (p < 0.001): a Fisher Pairwise comparison indicated that the magnitude of afore mentioned parameters for obese subjects was larger than those observed for other BMI groups. Post-hoc power analysis indicated power equal or larger than 95% when inferring statistical differences in VO2, VCO2, EER, and HR across BMI groups.

Moreover, simple linear regression models relating BMI to metabolic parameters were explored, and the results are reported in Fig 3. Significant positive relationships between

**Table 4. Summary of subjects' occupations.** All the data are reported as number of subjects, and standard occupational classifications are based on the U.S. Bureau of Labor Statistics [57].

| Standard occupational classification | Title | Number of subjects | Total |
|---|---|---|---|
| *Educational instruction and library* | Graduate student | 5 | 16 |
| | Research associate | 2 | |
| | Teacher | 1 | |
| | Undergraduate student | 8 | |
| *Healthcare practitioners and technical* | Nurse | 1 | 2 |
| | Physical therapist | 1 | |
| *Management, Sales and related* | Cashier | 4 | 6 |
| | Nonprofit manager | 1 | |
| | Sales | 1 | |
| *Service Industry* | Appraiser | 1 | 6 |
| | Driver | 1 | |
| | Engineer | 1 | |
| | Janitor | 1 | |
| | Line cook | 1 | |
| | Translator | 1 | |

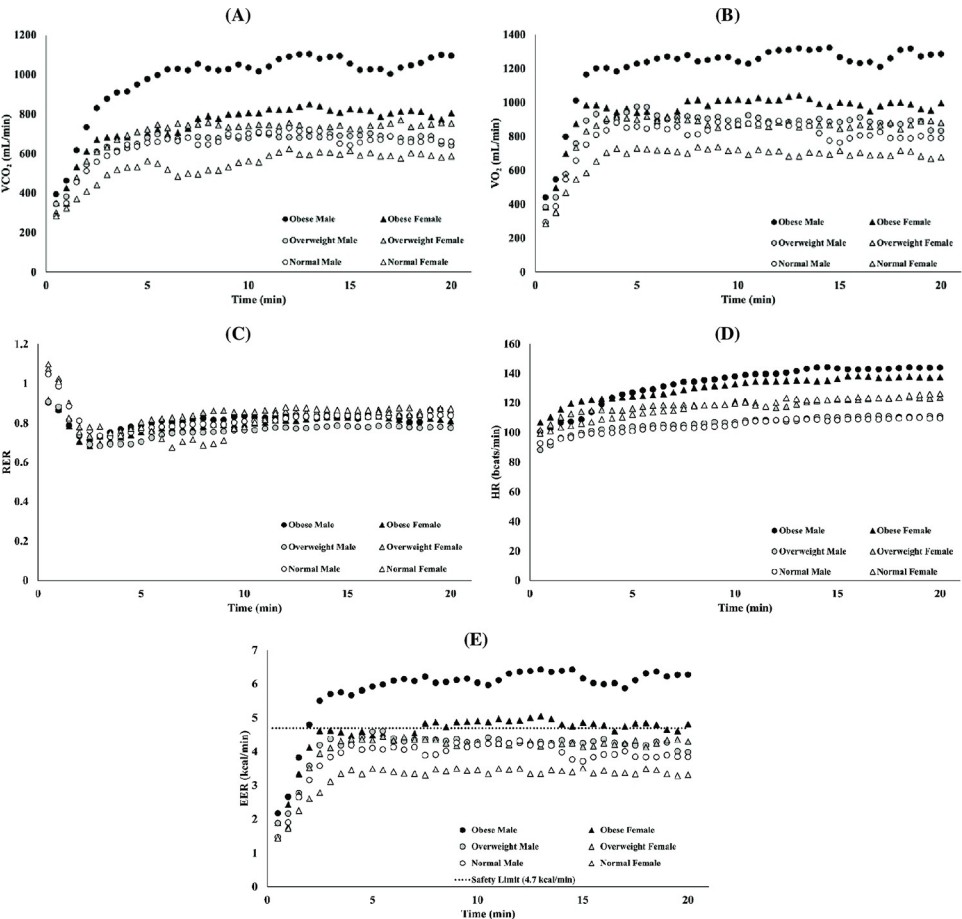

**Fig 2. Time-dependent metabolic parameters across testing period.** Subjects were grouped into their corresponding BMI classifications and gender: black circle is obese male, black triangle is obese female, grey circle is overweight male, grey triangle is overweight female, white circle is normal male, and white triangle is normal female. An average value for five metabolic dependent variables was taken for each BMI-gender group over the testing period. The figure represents the relationship between elapsed time and average: (A) VCO2; (B) VO2; (C) RER; (D) HR; and (E) EER, with the dotted line representing the safety limit (4.7 kcal/min).

subjects' BMI and VCO2 ($R^2$ = 59.65%, $p < 0.001$), VO2 ($R^2$ = 45.01%, $p < 0.001$), and HR ($R^2$ = 21.86%, $p = 0.009$) were found. This was not the case for RER ($R^2$ = 1.53%, $p = 0.515$). Importantly, BMI had a significant impact on EER ($R^2$ = 50.83%, $p < 0.001$), with 80% of obese subjects exceeding the NIOSH safety limits, see Fig 3E.

## 4. Discussion

Previous studies have identified a need to consider physiological implications in calculating RWL for all populations [29]. Barim et al. showed that the inclusion of new physiological RNLE multipliers, one of which was BMI, improved risk assessment of musculoskeletal injuries [30]. Still, the effect of BMI on metabolic parameters during repeated manual lifting tasks has not been quantified. To the authors' best knowledge, this is the first study measuring and comparing the effects of clinically elevated BMI on indicators of physiological stresses at RWL. The overall results of this study indicate that obese subjects exceeded the NIOSH safety limit for EER during this lifting task.

**Table 5. Metabolic variables versus BMI and gender.** Data are reported in terms the average value at the steady state of volume of carbon dioxide production (VCO2), volume of oxygen consumption (VO2), respiratory exchange ratio (RER), heart rate (HR), and energy expenditure rate (EER) in comparison to a subject's BMI classification and gender. Marginal means ± standard deviation for BMI classifications are grouped using a Fisher Pairwise comparison. A p-value for the ANOVA test comparing each metabolic variable with BMI class is also reported, with (*) indicating statistically significant difference.

| | | Normal (n = 10) | Overweight (n = 10) | Obese (n = 10) | Subtotal (n = 30) | P-Value |
|---|---|---|---|---|---|---|
| VCO2 (mL/min) | Male (n = 15) | 680.9 ± 42.3 | 684.5 ± 33.0 | 1060.3 ± 50.9 | *808.6 ± 183.0* | 0.000* |
| | Female (n = 15) | 593.3 ± 43.4 | 744.1 ± 38.8 | 814.6 ± 36.9 | *717.3 ± 100.5* | |
| | Subtotal (n = 30) | *637.1 ± 31.1* | *714.3 ± 26.1* | *937.4 ± 32.1* | *762.9 ± 130.8* | |
| | Grouping | B | B | A | | |
| VO2 (mL/min) | Male | 821.7 ± 58.4 | 886.9 ± 47.8 | 1279.5 ± 62.4 | *996.0 ± 210.0* | 0.000* |
| | Female | 695.7 ± 46.4 | 869.6 ± 48.7 | 998.1 ± 50.5 | *854.5 ± 133.1* | |
| | Subtotal | *758.7 ± 38.9* | *878.2 ± 36.0* | *1138.8 ± 39.6* | *925.3 ± 163.2* | |
| | Grouping | B | B | A | | |
| RER | Male | 0.83 ± 0.01 | 0.78 ± 0.01 | 0.83 ± 0.03 | *0.81 ± 0.02* | 0.679 |
| | Female | 0.86 ± 0.03 | 0.86 ± 0.01 | 0.82 ± 0.01 | *0.84 ± 0.03* | |
| | Subtotal | *0.84 ± 0.02* | *0.82 ± 0.01* | *0.82 ± 0.02* | *0.83 ± 0.03* | |
| | Grouping | A | A | A | | |
| HR (BPM) | Male | 109.6 ± 0.8 | 110.7 ± 0.9 | 143.3 ± 0.8 | *121.2 ± 15.7* | 0.012* |
| | Female | 124.5 ± 1.7 | 123.2 ± 1.2 | 137.7 ± 1.0 | *128.5 ± 6.7* | |
| | Subtotal | *117.0 ± 1.0* | *117.0 ± 0.8* | *140.5 ± 0.7* | *124.8 ± 11.1* | |
| | Grouping | B | B | A | | |
| EER (kcal/min) | Male | 4.1 ± 0.3 | 4.3 ± 0.2 | 6.3 ± 0.3 | *4.9 ± 1.0* | 0.000* |
| | Female | 3.4 ± 0.2 | 4.2 ± 0.2 | 4.9 ± 0.2 | *4.2 ± 0.6* | |
| | Subtotal | *3.8 ± 0.2* | *4.3 ± 0.2* | *5.6 ± 0.2* | *4.5 ± 0.8* | |
| | Grouping | B | B | A | | |

Steady state values of $VCO_2$ and $VO_2$ provide crucial information on respiratory gas exchange during exercise. In the present study, the magnitudes of these parameters were significantly dependent on BMI group ($p < 0.001$): $VCO_2$ values associated with obese subjects were 27.0% higher than overweight subjects and 38.1% higher than normal subjects; $VO_2$ values associated with obese subjects were 25.8% higher than overweight subjects and 40.0% higher than normal subjects. The higher values of $VCO_2$ and $VO_2$ for obese people in the present study may be due to several reasons. First, obese people have a higher cardiac stroke volume and a higher mechanical demand on their lungs due to increased weight bearing [31–33]. They also suffer from metabolic inefficiencies and in many cases a low grade systemic inflammation felt to create a higher request for oxygen during exercise [34, 35], resulting in increased respiratory rates for a given workload [36]. Finally, previous studies report that resting $VCO_2$ and $VO_2$ are naturally higher in people of clinically elevated BMI classified as obese [35, 37]. These gas volumes are shown to proportionally increase progressively from resting levels to intense exercise [35–37]. The increased inspiratory and expiratory gas volumes observed in obese subjects compensate for reduced cardiopulmonary efficiency and are associated with increased adipose tissue, and these effects are only exacerbated by exercise as the work of breathing is increased.

RER is another metabolic parameter that is used to relate $VCO_2$ and $VO_2$ when lifting at RWL. In the present study, all genders and BMI classifications performed similarly, achieving a steady state RER between 0.82 and 0.84. Thus, the increases in $VCO_2$ and $VO_2$ associated with higher BMI groups are uniform and result in similar RER values across BMI groups. This finding is in agreement with previous studies showing no relationship between exercise RER and absent body fat [38]. Furthermore, typical values for exercise RER vary between 0.7 and

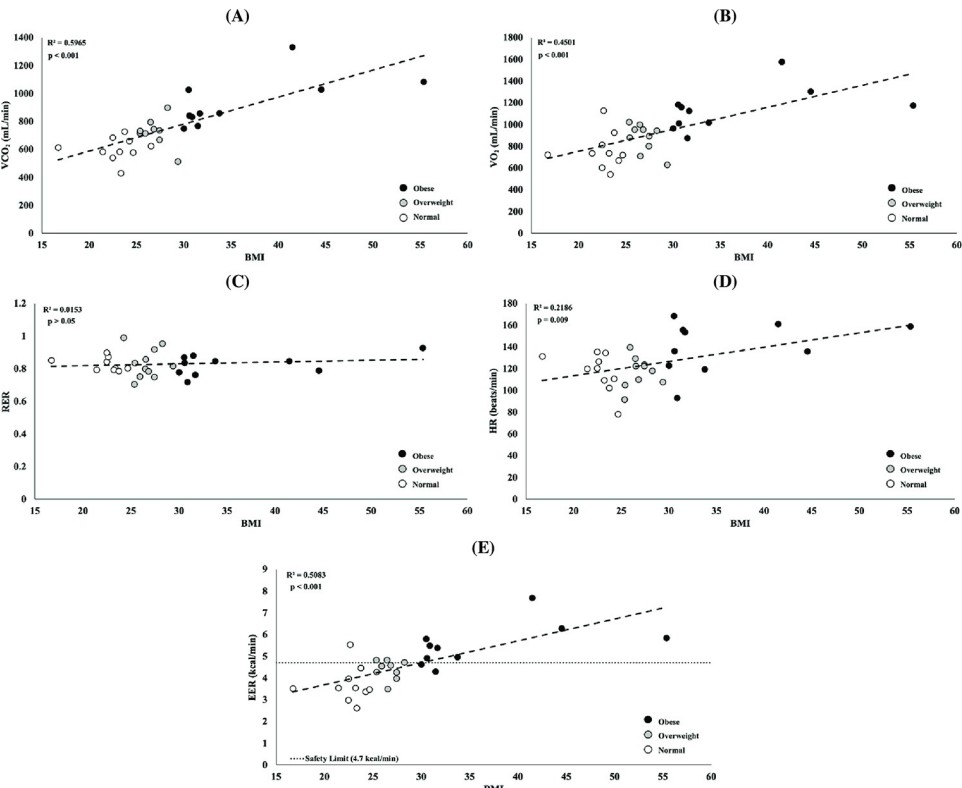

**Fig 3. Relationships between metabolic variables and BMI.** Each circle represents a subject, and the colors of the circles correspond to the BMI of that subject: black is obese, grey is overweight, and white is normal. An average steady state value for each metabolic variable was graphed versus each subject's BMI. Dashed lines reflect the linear regressions, and $R^2$ and p-values are provided for the data of each metabolic variable. The figure represents the relationship between a subject's BMI and average steady state: (A) VCO2; (B) VO2; (C) RER; (D) HR; and (E) EER, with the dotted line representing the safety limit (4.7 kcal/min).

1.0, which gives insight into the energy source used to replenish ATP [35, 37]. Values close to 0.7 reflect lipid oxidation (mainly during lower intensity exercise), while values closer to 1.0 reflect carbohydrate oxidation (mainly during higher intensity exercise) [35, 37]. Thus, all subjects in the present study relied upon both lipid and carbohydrate oxidation to perform lifting tasks. However, lipid oxidation was favored; this is a reasonable finding for a moderate intensity activity [35, 37].

In contrast to RER, the higher values of $VCO_2$ and $VO_2$ in the obese and overweight BMI groups resulted in a significant effect on EER. The obese group had a 26.3% higher EER than the overweight group and 38.3% higher EER than the normal group. This is likely because $VCO_2$ and $VO_2$ are linear factors in Weir's equation, see Eq 2 [39]. Previous studies reported that obese people have both a greater basal metabolic rate and physical activity cost than lower BMI people [40, 41]. DeLany et al. suggested that these differences were due to the obese population generally having biomechanical inefficiencies that cause them to work harder and thus have a higher metabolic cost [40, 42]. These metabolic inefficiencies, in turn, contribute to a higher oxygen demand for a given workload seen in clinically elevated high BMI individuals during exercise [34, 35]. Similar to the components that contribute to $VCO_2$ and $VO_2$, exercise EER is likely increased for the higher BMI groups because of the larger oxygen demand and inherent inefficiencies associated with increased body weight.

Furthermore, Fig 2C depicts the differences in EER between BMI groups in reference to the NIOSH safety limit. The EER of the obese group (5.6 kcal/min) surpassed the safety threshold

(4.7 kcal/min), but this was not the case for overweight (4.3 kcal/min) and normal (3.8 kcal/min) groups. Specifically, Fig 3E shows that 80% of the tested obese subjects exceeded the NIOSH safety limit for EER during this lifting task. Overall, the relationship between EER and BMI confirms our hypothesis: for clinically elevated BMI individuals, manual lifting at RWL produced a level of physiological stress above safety limits established by the RNLE.

Similar to the EER, BMI class also significantly impacted HR, see Table 5. Obese subjects experienced higher HRs than both overweight and normal subjects. Notably, the obese subjects had a 20% higher HR than the other two BMI groups. Previous studies reported that obese people have an elevated resting HR [43–46], which puts them at increased risk for cardiometabolic problems [47–49]. Cardiorespiratory fitness, which is more common among people of lower BMI, has been associated with a decrease in resting heart rate [50, 51]. Moreover, heart rate increases from resting to exercise, and is linearly dependent on exercise intensity [52, 53]. These established findings explain the increased HR observed in subjects of higher BMI during the exercise activity in the present study.

To further explore the relationships between BMI and metabolism, simple linear regressions were explored, see Fig 3. There was a positive correlation between a subjects' BMI and $VCO_2$ ($R^2$ = 45.01%; p < 0.001), $VO_2$ ($R^2$ = 59.65%; p < 0.001), EER ($R^2$ = 50.83%; p < 0.001), and HR ($R^2$ = 21.86%; p = 0.009). However, BMI did not significantly influence RER ($R^2$ = 1.53%; p = 0.515). These results corroborate our findings on the effect of BMI on metabolic parameters (Table 5).

Gender also had some effect on certain metabolic variables. Males had 15.3% higher $VO_2$ (p = 0.014) and 15.4% higher EER (p = 0.017) than females. This relationship is in agreement with previous studies, which showed that men had a significantly greater $VO_2$ [54, 55] and EER [55, 56] than women during steady-state submaximal exercise. Thus, there are likely underlying physiological differences between males and females that are evidenced through $VO_2$ and EER values.

Some limitations should be acknowledged. A total of 30 subjects (15 male and 15 female) were evaluated in the present study. While a larger pool of participants would be desirable, results from the post-hoc power analysis showed that the enrolled pool was large enough to observe effects of BMI (n = 10) and gender (n = 15) with power equal or larger than 95%. Also, our subjects had occupations that did not necessarily involve manual material handling, see Table 4. Future studies should be conducted to investigate the effects of physiological stress on subjects whose occupation specifically involves manual material handling to draw more definitive conclusions about risk for occupational injury. In addition, BMI does not take into account body fat percentage or distribution. While the researchers only selected obese participants with little to no reported exercise history, body fat could still be an influential variable that should be analyzed in the future. Furthermore, the repetitive lifting activity performed in this analysis only represents a single component of the many manual material handling tasks. Future effort should be devoted to investigating additional tasks that comprehensively reflect the typical manual activities performed in workplace environment. Still, the activity we used was conducted according to NIOSH guidelines to enforce physiological safety limits. Thus, the present study can be used as a basis for understanding how clinically elevated BMI may impact a worker's risk for injury.

In summary, this study examined the effects of BMI on physiological stress during a manual lifting task designed according to the established safety guidelines of the RNLE. It was found that elevated BMI led to an increase in $VCO_2$, $VO_2$, EER, and HR. Of note, obese subjects by BMI criteria exceeded the NIOSH energy expenditure safety limit, and suggests that these individuals would be at increased risk for workplace injury as compared to their overweight and normal counterparts. The results of this study can be used to determine additional BMI

multipliers for the RNLE to help reduce the risk of physiological stress, potentially leading to injury, for workers with a larger BMI. Future research should focus on larger populations for the purpose of establishing cut-points for increased BMI.

## Supporting information

**S1 File.**
(XLSX)

## Author Contributions

**Conceptualization:** Eduard Tiozzo, Thomas M. Best, Francesco Travascio.

**Data curation:** Sergio A. Lemus, Mallory Volz, Francesco Travascio.

**Formal analysis:** Sergio A. Lemus, Mallory Volz, Francesco Travascio.

**Funding acquisition:** Eduard Tiozzo, Francesco Travascio.

**Investigation:** Sergio A. Lemus, Mallory Volz, Eduard Tiozzo, Arlette Perry, Thomas M. Best, Francesco Travascio.

**Methodology:** Sergio A. Lemus, Mallory Volz, Francesco Travascio.

**Project administration:** Sergio A. Lemus.

**Supervision:** Mallory Volz, Francesco Travascio.

**Writing – original draft:** Sergio A. Lemus, Mallory Volz, Eduard Tiozzo, Arlette Perry, Thomas M. Best, Francesco Travascio.

**Writing – review & editing:** Sergio A. Lemus, Mallory Volz, Eduard Tiozzo, Arlette Perry, Thomas M. Best, Francesco Travascio.

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
