## [Decision Letter · Decision Letter 0]

21 Nov 2022

PONE-D-22-28339THE EFFECT OF CLINICALLY ELEVATED BODY MASS INDEX ON PHYSIOLOGICAL STRESS DURING MANUAL LIFTING ACTIVITIESPLOS ONE

Dear Dr. Travascio,

Thank you for submitting your manuscript to PLOS ONE. After careful consideration, we feel that it has merit but does not fully meet PLOS ONE’s publication criteria as it currently stands. Therefore, we invite you to submit a revised version of the manuscript that addresses the points raised during the review process.

 Please address all issues raised by the reviewers particularly on the manuscript being misleading: "My honest concern is the small sample size, as the design of the study recruiting volunteers of different subject groups to TWU workforce, I feel that the article is misleading and unable to represent actual target population."

We look forward to receiving your revised manuscript.

Kind regards,

Shazlin Shaharudin

Academic Editor

PLOS ONE

Journal Requirements:

"The project described was supported by Grant Number UM-PRA-2022-3243 from the University of Miami."

"FT and ET received funds from The University of Miami (award UM-PRA-2022-3243) to complete this study. The funders had no role in study design, data collection and analysis, decision to publish, or preparation of the manuscript."

Additional Editor Comments:

Dear authors,

Please address this major concern by a reviewer: My honest concern is the small sample size, as the design of the study recruiting volunteers of different subject groups to TWU workforce, I feel that the article is misleading and unable to represent actual target population.

Reviewers' comments:

Reviewer's Responses to Questions

**Comments to the Author**

1. Is the manuscript technically sound, and do the data support the conclusions?

Reviewer #1: Partly

Reviewer #2: Yes

2. Has the statistical analysis been performed appropriately and rigorously? 

Reviewer #1: No

Reviewer #2: Yes

3. Have the authors made all data underlying the findings in their manuscript fully available?

Reviewer #1: Yes

Reviewer #2: Yes

4. Is the manuscript presented in an intelligible fashion and written in standard English?

Reviewer #1: Yes

Reviewer #2: Yes

5. Review Comments to the Author

Reviewer #1: Comments

No Section Comment

1. Abstract

- NIOSH – please give full abbreviation

2. Methodology

- Aren’t this experimental study?

- Sample size calculation? Cross sectional study but involving 30 subjects only? Although authors justified the matter in limitation of study that sample size N=30 was enough to show effects, however when compare between the three groups of normal, overweight and obese, the sample size = 10 only.

- Inclusion and exclusion criteria?

- Pls indicate in the article that this study is granted ethical approval

3. Results

- Table 1: In text citation mentioned as Table 1, but the Table were presented as Table 1A, 1B and 1C as we know that it should tally. Suggest to either change in text citation to be specific Table 1A/1B/1C or if to use ‘Table 1’ hence combine all three tables (Table 1A/1B/1C) into 1 table only

- Table 3: please include note for each abbreviation to be written in full term. The results presented as average/mean, please include also the standard deviation

- Figure 3 (A), (B), (C), (D) and (E) were not properly labelled

4. Discussion

- Although the authors already mentioned in the study limitation that subjects selected do not belong to TWU workforce, yet such focus on TWU only briefly stated in introduction; but not comprehensively discussed in the discussion. It’s a given that the risk factors that the paper looked at were both on sociodemographic (gender and BMI), however as the title mentioned on manual lifting and scope on TWU, as well as referring to NIOSH guidelines, it’s kind of misleading as it’s not the occupational factors that being discussed. Moreover, only an aspect of TWU was looked at.

- Suggest to tune the paper towards general population then, if not able to thoroughly relate/focus to occupational setting.

Thank you.

Reviewer #2: Dear Authors, it is my pleasure to be able to review this manuscript. This is an interesting study, and I hope that the said organization can implement your suggestion to include BMI as one of the variable/denominator in the RNLE. There were a few comments to improve this manuscript, which you can find in the attached PDF. One of it was to further elaborate the sampling of the respondents, on how do the respondents replied to the advertisement, and how do you select the volunteer. Were there only 31 individuals who first interested to participate in the study? Or if there were many of them, how would you counter selected them after considering the exclusion criteria? I think this part was not clear and need further elaboration. You should also used S.I unit for any measurement described in the manuscript. Such as the BMI, or weight of the load. The tables and figures in the manuscript should also be quoted according to the sequence mentioned in the text, for easy referencing and understanding of the reader. Other than these three major things to be pointed out, you are doing good! Please find other tiny comments as described in the PDF. All the best!

6. PLOS authors have the option to publish the peer review history of their article (what does this mean?). If published, this will include your full peer review and any attached files.

Reviewer #1: No

Reviewer #2: No

---

## [Author Response · Author response to Decision Letter 0]

23 Nov 2022

Responses to Reviewers’ Comments

We thank the Editor and the Reviewers for this opportunity to improve the quality of our contribution. We have taken in the most serious consideration the suggestions of the Reviewers and addressed their concerns as it follows below. Specifically, additional clarifications have been provided on the enrollment procedures adopted to recruit human subjects. The naming of the tables has been changed as per suggestions of both Reviewers and their content has been enriched with clarification and additional data (standard deviations of the marginal mean values reported in Table 3, now Table 5). Additional elaborations have been included in the Discussion, as per request of Reviewer #1. Revisions to the manuscript are highlighted in yellow.

Reviewer #1

1.Abstract

-NIOSH – please give full abbreviation

Response: done as requested.

2. Methodology

- Aren’t this experimental study?

Response: Yes, this was an experimental study involving human subjects. The details on recruitment and ethical approval are reported in section 2.1 of Methods.

- Sample size calculation? Cross sectional study but involving 30 subjects only? Although authors justified the matter in limitation of study that sample size N=30 was enough to show effects, however when compare between the three groups of normal, overweight and obese, the sample size = 10 only.

Response: We apologize for the confusion we may have created with he language used in the limitations of the study (9th paragraph of discussion). In fact, while the total number of participants was 30, the actual sample size was 10 (when comparing body masses). Our post-hoc analysis, based on n = 10 (not 30), indicated that even with this sample size the power was larger than 95%. In order to clarify this, the language in the text was modified. It now reads: “A total of 30 subjects (15 male and 15 female) was used in the present study. While a larger pool of participants would be desirable, results from the post-hoc power analysis show that the enrolled sample was large enough to observe effects of BMI (n = 10) and gender (n = 15) with power equal or larger than 95%.”.

- Inclusion and exclusion criteria?

Response: The criteria have been detailed in section 2.1 of Methods. It reads: “Inclusion criteria included any adult with no current or prior history of musculoskeletal injury, or any medical condition preventing him/her from performing the physical activity planned in the experiments. Exclusion criteria for the study consisted of current pregnancy, history of musculoskeletal injuries in the past 6 months or any chronic cardiopulmonary-related conditions that may be exacerbated during tests (including but not limited to asthma, chronic obstructive pulmonary disease, congestive heart failure, and/or lung disease).”

- Pls indicate in the article that this study is granted ethical approval

Response: Ethical approval was granted by the IRB of the University of Miami. This has been specified at the beginning of section 2.1 of Methods. It reads: “The methods used in this cross-sectional study were approved by the Internal Review Board of the University of Miami (IRB ID: 20211175).”

3. Results

- Table 1: In text citation mentioned as Table 1, but the Table were presented as Table 1A, 1B and 1C as we know that it should tally. Suggest to either change in text citation to be specific Table 1A/1B/1C or if to use ‘Table 1’ hence combine all three tables (Table 1A/1B/1C) into 1 table only

Response: Done as suggested. All the data have been reported in 3 distinct table: Table 2 (previously Table 1A), Table 3 (previously Table 1B) and Table 3 (previously Table 1C).

- Table 3: please include note for each abbreviation to be written in full term. The results presented as average/mean, please include also the standard deviation

Response: Table 3 is now Table 5. The description of the abbreviations has been added in the table caption. In addition, data are now reported as mean ± standard deviation, as requested. 

- Figure 3 (A), (B), (C), (D) and (E) were not properly labelled

Response: Upon building the PDF from the files uploaded on the website, the labeling of figure 3 was wrong. We apologize for the inconvenience. The labeling has now been fixed.

4. Discussion

- Although the authors already mentioned in the study limitation that subjects selected do not belong to TWU workforce, yet such focus on TWU only briefly stated in introduction; but not comprehensively discussed in the discussion. It’s a given that the risk factors that the paper looked at were both on sociodemographic (gender and BMI), however as the title mentioned on manual lifting and scope on TWU, as well as referring to NIOSH guidelines, it’s kind of misleading as it’s not the occupational factors that being discussed. Moreover, only an aspect of TWU was looked at.

Response: We agree with the Reviewer’s point. The issue of TWU workers is only marginally treated in the introduction of the paper. In this revised manuscript,we eliminated any specific reference of this work to the TWU sector as the implications of this study have applicability for any occupation involving manual material handling. Pertinent changes in the manuscript have been made at the beginning of the Abstract, in the first two paragraphs of the Introduction, and the in the 9th paragraph of Discussion.

- Suggest to tune the paper towards general population then, if not able to thoroughly relate/focus to occupational setting.

Response: Done as suggested. See response to previous comment.

Reviewer #2

Dear Authors, it is my pleasure to be able to review this manuscript. This is an interesting study, and I hope that the said organization can implement your suggestion to include BMI as one of the variable/denominator in the RNLE. There were a few comments to improve this manuscript, which you can find in the attached PDF. 

- please include ethics certificate number

Response: done as requested. See section 2.1.

- What was the type of sampling used to sample the respondent? Were the respondents selected based on their BMI and other exclusion criteria? What was the response rate? Was there any age range? I think this part should be described in detail.

Response: The respondents replied to announcements described in a flyer distributed in common areas of the University of Miami and local surroundings. Social media (Facebook) were also used to advertise the study. All subjects were selected according to our inclusion and exclusion criteria. Specifically, inclusion criteria consisted of any adult with no current or prior history of musculoskeletal injury, or any medical condition preventing him/her from performing the physical activity planned in the experiments. Exclusion criteria for the study include current pregnancy, history of musculoskeletal injuries in the past 6 months, or any chronic cardiopulmonary-related conditions that may be exacerbated during testing (including but not limited to asthma, chronic obstructive pulmonary disease, congestive heart failure, and/or lung disease). Due to budget constraints, we aimed at 30 individuals (15 males and 15 females) evenly distributed across 3 BMI groups (normal, overweight, and obese). We stopped our enrollment as we reached the cap of 30 individuals, fortunately all 30 subjects completed the study. All this information is reported in the revised version of the manuscript. See section 2.1 of Methods. 

- Table 2. This should be quoted as Table 1, since it was the first table mentioned in the article

Response: done as requested. Table 2 is now Table 1.

- All of the tables should be quoted according to the sequence in the text

Response: done as requested. All of the tables are now labeled according to the sequence they appear in the text.

- Table 2. I think the author should used S.I. unit for easy understanding of the readers.

Response: We agree that the S.I. unit system is universally understood in the scientific community and, in general, it’s use would increase the clarity of the information reported in a scientific publication. However, the information reported in Table 2 refers to the values of the RNLE coefficients used in this study. Traditionally, this information is reported in Imperial units. In conformity with the literature in the field, we decided to report all information in inches and pounds. We believe that this approach improves the clarity of the experimental conditions adopted in this study.

---

## [Editor Report · Decision Letter 1]

28 Nov 2022

THE EFFECT OF CLINICALLY ELEVATED BODY MASS INDEX ON PHYSIOLOGICAL STRESS DURING MANUAL LIFTING ACTIVITIES

PONE-D-22-28339R1

Dear Dr. Travascio,

We’re pleased to inform you that your manuscript has been judged scientifically suitable for publication and will be formally accepted for publication once it meets all outstanding technical requirements.

Kind regards,

Shazlin Shaharudin

Academic Editor

PLOS ONE
---

## [Editor Report · Acceptance letter]

1 Dec 2022

PONE-D-22-28339R1 

The effect of clinically elevated body mass index on physiological stress during manual lifting activities 

Dear Dr. Travascio:

I'm pleased to inform you that your manuscript has been deemed suitable for publication in PLOS ONE. Congratulations! Your manuscript is now with our production department. 

Kind regards, 

on behalf of

Dr. Shazlin Shaharudin 

Academic Editor

PLOS ONE